# Elicitation Promoability with Gamma Irradiation, Chitosan and Yeast to Perform Sustainable and Inclusive Development for Marjoram under Organic Agriculture

Tarek E. Sayed * and El-Sayed S. Ahmed

Radioisotope Department, Egyptian Atomic Energy Authority, Cairo 11787, Egypt
* Correspondence: dr.tarekelsayed64@gmail.com

**Abstract:** Sweet marjoram (*Majorana hortensis*) is an important aromatic herbal plant that has long been used and well managed in the traditional and general medical, pharmaceutical, food, cosmetic, and perfume industries. Thus, the increase in its productivity appears to be of great value since there is a large number of bioactive secondary metabolites as well as an increase in the demand in domestic or foreign markets. The purpose of this study is the possibility of promoting the sustainable development of marjoram in the framework of organic farming through gamma irradiation, chitosan and yeast. Field experiments were conducted in a factorial split-plot design with three iterations over two consecutive seasons (2019 and 2020). The main plot is an abiotic elicitor (15 Gy gamma irradiation), two biotic elicitors 500 ppm chitosan, 0.5% yeast, and a non-elicitor (as control), while in the sub-main plot, there were two organic fertilizers, water extract of moringa 20 g/m$^2$ dry leaves, 20 g/m$^2$ fulvic acid, and 20 g/m$^2$ (NPK); the latter is a traditional agrochemical. Statistical analysis of all characteristics of production and quality of biomass and biologically active secondary metabolites revealed that the use of organic fertilizers helped in increasing the yield of marjoram, both qualitatively and quantitatively, and significantly outperformed the chemical fertilizer. The experiment enhances the comprehensive and integrated development of marjoram under organic cultivation and achieves a promising alternative to traditional cultivation without the use of microbicides and/or agrochemical pesticides.

**Keywords:** elicitation; gamma irradiation; organic agriculture; secondary metabolites; yeast

## 1. Introduction

Sustainability is currently a global issue that has provoked significant challenges for modern economic technologies and researchers, especially in using plants to solve major environmental problems worldwide [1,2]. With great application of nuclear technologies, the risk for humans and the environment has been considered of significant importance. In concert with sustainability and nuclear applications, green and dry plants [3,4] have an important role in the remediation of wastewater contaminated with stable and radioisotopes then followed by sustainable stabilization in cement [5,6]. Moreover, plants could be used in treatment such as aiding the recovery of the workers at the Chernobyl accident site using a daily oral dose of G. Biloba [7]. Medicinal plants and their constituents can alter radiolabeling and biodistribution via several mechanisms [8]. On the other hand, nuclear radiation can be used to improve the efficiency and performance of medicinal plants used in modern and traditional medicinal applications [9].

Sweet marjoram (*Majorana hortensis*) is a precious herbaceous aromatic plant native to the Mediterranean region that belongs to the *Lamiaceae* family [10]. In addition, it is a medicinal plant that is widely applied in domestic medical systems. Sweet marjoram is grown in numerous Asian, North African, and European nations, including Tunisia, Egypt, India, France, Hungary, Germany, Spain, Hungary, Portugal, Poland, and France [11].

Egypt has been internationally known for the cultivation and export of quality marjoram fresh or dry tender stems and the leaves are used for condiments and spices [12].

The leaves and flowers of the plant contain delicate fragrant essential oils that are widely used in traditional medicinal including, but not limited to, lotions, perfumes, creams, and soap [13]. The use of the plant has been documented to treat many ailments such as headaches, asthma, ear problems, and others. Additionally, cramps, depression, dizziness, gastrointestinal disorders, migraine, headache, paroxysmal cough, and a diuretic are all treated with *O. majorana*. Its essential oil is used in perfumery, the pharmaceutical industry, and cosmetics.

Recent studies have shown the vital role of marjoram in the treatment of cancer and as an active anti-fungal, antioxidant and cytotoxic agent. The antioxidant properties of marjoram have been studied extensively [14]. However, it is used mainly as a spice and natural preservative for foods, especially meat.

Elicitors are described as herbal or synthetic (organic or non-biological) substances, and while they are permeable to plants at low levels, they stimulate stress responses in plants and aid in the synthesis of secondary metabolites [15]. Elicitors serve as signals, and elicitation begins with signal perception by eliciting specific receptors on the plant cell membrane, which is followed by the initiation of a signal transduction cascade, which subsequently changes the expression levels of different governance transcription genes of secondary metabolic pathways [16,17]. Elicitation is a successful and commonly used biotechnological method for inducing new secondary metabolites in plants [18].

Elicitors' innate plant resistance mechanisms are induced using biotic and abiotic factors. Elicitation can be an important strategy for producing bioactive secondary metabolites (BSMs). It has been suggested that foliar application of these compounds' (abiotic & biotic elicitors) under normal and stressful conditions improved phytochemicals and biosynthetic pathways of secondary metabolite (BSM) content in plants, including medicinal and aromatic plants [18,19].

Elicitors improve the quality and quantity of BSMs that promote health [17,19,20]. Changes in elicitor growth and development can affect morphological, physiological, and biochemical properties and improve biomass production and quality [21,22].

Attention to agriculture has become imperative over time and implementation of sustainability in this field is crucial. To move forward towards sustainability involves maintaining modern technology and rationing the use of pesticides of various kinds to increase production. These pesticides have been selectively used on medicinal and aromatic plants as well as cereals and horticulture. Pesticides have a negative impact on the environment and hinder long-term growth [23], and several studies have linked the increased use of these pesticides to health issues [24,25]. Organic farming has grown in popularity in recent years, as has pharmaceutical production of organic medicinal and aromatic plants as an alternative to regular agriculture [26]. Considering the above facts according to recent trends and future prospects of various strategies to direct increased productivity of biomass and bioactive secondary metabolites in medicinal and aromatic plants are highlighted. It is notwithstanding works dealing with these aspects are scarce. Hence, the present article highlights elicitation coupled with organic fertilizer mediating biomass and bioactive secondary metabolites production and quality to promote the achievement of sustainable development for marjoram under organic agriculture.

## 2. Materials and Methods

### 2.1. Practical Field Experiment

Two consecutive seasons of field experiment trials, 2019 and 2020, were constructed as split-plot design with three replications based on randomized full block design.

The primary plot had three elicitors: 15 G gamma irradiation (GI), 500 ppm chitosan (CH), and 0.5 g/L (YS) solution in water with 0.1 percent ($v/v$) Tween 20 as a surfactant. The sub-main plot contained three fertilizers, two organic fertilizers, Moringa dry leaves

water- extract 20 g/m$^2$ (MO), fulvic acid 20 g/m$^2$ (FA), and traditional chemical fertilizer NPK, 20 g/m$^2$.

Gamma irradiated and non-irradiated marjoram seeds on 1st February for both seasons were planted in trays containing soil, sand, beat mixed (1:1:1) ratio (*v/v*) subsequently established in a greenhouse. Seedlings 4 weeks of age were transplanted to the field in plots 4 × 4 m in row 50 cm interspacing that contain 64 plants/plot. At such seasons, plants' leaves were sprayed with water, (Zero elicitor), (NA)CH, and YS a non-elicitor or solution two times 1st April, 1st May before 1st harvesting on 1st June. They were also sprayed on 1st July and 1st August before harvesting on 1st September.

### 2.2. Biomass Yield Production

The fresh weight of leaves was determined pre-plot and per m$^2$ for the 2019 and 2020 seasons.

### 2.3. Bioactive Secondary Metabolites Production (BSMs)

### 2.3.1. Phenolics (TPC)

The Folin–Ciocalteu reagent was used to determine total phenol content, as described in [27]. A diluted sample extract (1 mg/mL) aliquot was mixed with 125 L of Folin-Clocaiteu reagent and 500 L of water. The mixture was stirred and left for 5 min before adding 125 g of 7% of Na$_2$CO$_3$. A 1 mL quantity of distilled water was added and well mixed. After 90 min in the dark, the absorbance at 760 nm was compared to a blank. Using a gallic acid solution, a calibration curve was constructed. Equations drawn from the standard gallic acid diagram are used to create content.

### 2.3.2. Flavonoids (TFC)

Flavonoid content was determined according to [28]. Sechium edule (jacq.) antioxidant activity of Swartz extract. Food Chemistry, 97:452:458. Each extract was dissolved in 1 mL of ethanol. One mL of the extract was combined with 1 mL of a 2% methanolic AlCl$_3$ solution. After 15 min of incubation at room temperature, the absorbance at 430 nm was measured, and the total flavonoids in the sample extract were quantified using a calibration curve (quercetin).

### 2.3.3. Flavonols (TFL)

The total flavonoids in *O. majorana* ethanolic extracts were calculated using the method described by [29]. Two mL AlCl$_3$ (2%) ethanol and 3 mL of sodium acetate solution (50 g/L) were mixed together. The blend was mixed before being incubated at 20 °C for 2 h. At 440 nm, the absorbance was measured. The amount of total flavonoids was measured in mg of quercetin equivalents per gram of dry weight (mg EQ/g DW).

### 2.3.4. Tannins (TAE)

Total tannins were calculated using a procedure devised by [30] based on the protein bovine serum albumin precipitation. The approach works by measuring the absorption of colored compound Fe$^{2+}$ phenols at 510 nm using spectrophotometry. The results were calculated in milligrams of tannic acid equivalent per gram of dry weight (mg TA/g DW).

### 2.3.5. Essential Oils (EO)

Fresh leaf samples from the plots were subjected to hydrodistillation for three hours to obtain content. The extracted essential oil was concentrated using sodium sulphate (anhydrous Na$_2$SO$_4$) and reserved in an amber glass sealed with Teflon kept at 4 °C even analysis as the following:

1-EO% = Extracted EO, g/ground leaves sample X100-moisture

2-EOY, Kg/m$^2$ = EO% X FLY, Kg/m$^2$ EO

Essential oil contents (EOC), analyzed by GC/MS using Shimadzu-HPLC System (Shim-adze Co., Kyoto, Japan) prepared with a CBM–20 AH controller, and LC20 AP Pump and an SPD–M$_2$OA Photo Diode Array (FDA) detector.

## 3. Statistical Analysis

Using a statistical analysis method, the data sets were first checked for normality using the Anderson and Darling normality tests (SAS, 2003). There are no statistically significant differences in the data from the two seasons. As a result, the pooled mean values of two seasons for all attributes examined were statistically analyzed. LSD at the 1% level was used to compare the significant means.

## 4. Results and Discussion

Statistical analysis revealed that solitary application with elicitors and fertilizers achieved a significant positive impact for such lasted trait as, chitosan (CH) > yeast (YS) > γ- irradiation (GI) under Moringa (MO) > fulvic acid (FA) > NPK for both two seasons However, the drenches between the two seasons were insignificant.

### 4.1. Biomass Yield Production

Pooled fresh leaves yield (FLY), 6.060 KG/m$^2$ as control (NE.NPK), was increased significantly, expressed as % of control (0) as the flowing:

CH.MO, CH.FA, and CH.NPK were 27, 22, and 19 respectively. In addition, YS.MO, YS.FA, YS.NPK were 24, 21, and 17 respectively and GI.MO, GI.FA, GI.NPK were 20, 17, and 11 respectively. Additionally, NE.MO, NE.FA, NE.NPK were 11, 8, and 0 (control 6.060 Kg/m$^2$) respectively as listed in Table 1 and represented in Figure 1.

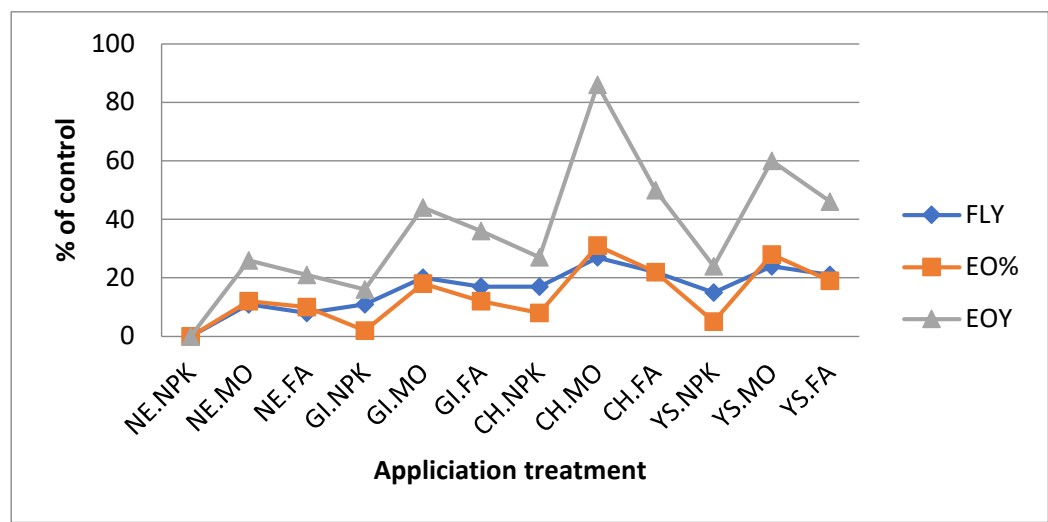

**Figure 1.** FLY, EO%, EOY.

### 4.2. Essential oils % (EO%)

PM EO %, 0.532% (control NE Significant, as % of control according to the following: CH.MO, CH.FA, CH.NPK were 3122, and 8, respectively. As well as, YS.MO, YS.FA, YS.NPK were 28, 19, and 5 respectively, and GI.MO, GI.FA, GI.NPK were 18, 12, and 2 respectively. In addition, NE.MO, NE.FA, and NE.NPK were 12, 10, and 0 (control 0.532 Kg/m$^2$) respectively as represented in Table 1 and Figure 1.

**Table 1.** Marjoram; fresh leaves yield, Kg/m$^2$, essential oil % and essential yield oil, Kg/m$^2$, in response to elicitation with gamma irradiation (GI), chitosan (CH) and yeast (YS) under the use of chemical fertilizer (NPK) and organic fertilizer; Moringa (MO) and fulvic acid (FA) over 2 subsequent seasons (2019 and 2020).

| Application Treatment | Fresh Leaves Yield, Kg/m$^2$ | | | Essential Oil % | | | Essential Oil Yield, Kg/m$^2$ | | |
|---|---|---|---|---|---|---|---|---|---|
| | 2019 | 2020 | PM | 2019 | 2020 | PM | 2019 | 2020 | PM |
| NE.NPK | 5.920 | 6.200 | 6.060(0) | 0.522 | 0.524 | 0.532(0) | 3.090 | 3.294 | 3.192(0) |
| NE.MO | 6.630 | 6.820 | 6.725(11) | 0.600 | 0.592 | 0.596(12) | 3.979 | 4.038 | 4.009(26) |
| NE.FA | 6.453 | 6.696 | 6.574(8) | 0.590 | 0.582 | 0.586(10) | 3.807 | 3.897 | 3.852(21) |
| GI.NPK | 6.690 | 6.882 | 6.786(11) | 0.548 | 0.540 | 0.544(2) | 3.666 | 3.716 | 3.691(16) |
| GI.MO | 7.163 | 7.440 | 7.301(20) | 0.632 | 0.629 | 0.630(18) | 4.527 | 4.680 | 4.604(44) |
| GI.FA | 7.045 | 7.254 | 7.149(17) | 0.616 | 0.603 | 0.609(12) | 4.340 | 4.374 | 4.357(36) |
| CH.NPK | 6.926 | 7. 228 | 7.228(19) | 0.580 | 0.571 | 0.575(8) | 4.017 | 4.071 | 4.044(27) |
| CH.MO | 7.637 | 7.750 | 7.693(27) | 0.705 | 0.692 | 0.699(31) | 5.385 | 5.363 | 5.374(68) |
| CH.FA | 7.282 | 7.502 | 7.392(22) | 0.653 | 0.645 | 0.649(22) | 4.744 | 4.839 | 4.792(50) |
| YS.NPK | 6.808 | 7.316 | 7.062(17) | 0.564 | 0.556 | 0.560(5) | 3.840 | 4.068 | 3.954(24) |
| YS.MO | 7.518 | 7.564 | 7.541(24) | 0.679 | 0.676 | 0.678(28) | 5.105 | 5.113 | 5.109(60) |
| YS.FA | 7.400 | 7.378 | 7.389(21) | 0.642 | 0. 624 | 0.633(19) | 4.751 | 4.604 | 4.678(46) |
| LSD1% | 0.035 | 0.031 | 0.027 | 0.003 | 0.005 | 0.004 | 0.022 | 0.025 | 0.026 |

Values between parentheses ware % over control. PM: pooled mean for two seasons (2019–2020).

*4.3. Essential Oils Yield (EOY, Kg/m$^2$)*

EOY, 3.192 Kg/m$^2$ (control) increased significantly as % of control as the following: CH.MO, CH.FA, CH.NPK, 68, 50, 27, respectively, and YS.MO, YS.FA, and YS.NPK, 60, 46 and 24, respectively.

Furthermore, GI.MO, GI.FA and GI.NPK were 4436,and 16, respectively. In addition, NE.MO, NE.FA, and NE.NPK were 26, 21, and 0 (control 3.192 Kg/m$^2$), respectively as represented in Table 1 and Figure 1. Biomass, FLY, Kg/m$^2$, EO%, EOC and Kg/m$^2$ were significantly increased in response to elicitor, CH > YS > GI integrated with MO > FA > NPK, despite the fact that no pest infestation and microbial disease incidences in the field experiment were detected in either season without using any agrochemical pesticides and/or micro-biocides. The disappearance of any pests and microbial diseases was in agreement with [31,32].) In addition, many researchers supported our results and declared a positive, significant impact on biomass yield production [33,34].

*4.4. Quali–Quantitative Bioactive Secondary Metabolites (BSMs)*

4.4.1. Essential Oil Contents (EOC)

EOC, 11 tirpenoids; &-terpnene, cineol, y-Terpinene, p-cymene, Terpineolene, D-linnalol, Terpineol-4ol, Bcaryphellene, &-Terpineol, Tymol, carvacol (Table 2) were measured. The total percentages of these 11 terpenoids were increased significantly as the following:

Values of CH.MO, CH.FA, and CH.NPK were 79.81, 76.48, and 69.66, respectively. Furthermore, YS.MO, YS.FA, and YS.NPK were 70.94, 73.80, and 71.32 respectively, and GI.MO, GI.FA, and GI.NPK were 71.60, 71.04, and 69.58 respectively. In addition, NE.MO, NE.FA, and NE.NPK were 69.03, 68.75, and 67.43 (control).

**Table 2.** Marjoram essential oil contents; in response elicitation with gamma irradiation (GI), chitosan (CH) and yeast (YS) under the use of chemical fertilizer (NPK) and bioorganic fertilizer; Moringa (MO)and fulvic acid (FA) in 2 subsequent seasons (2019 and 2020).

| Application Treatmen | Essential Oil Content from Terpinoids | | | | | | | | | | | |
|---|---|---|---|---|---|---|---|---|---|---|---|---|
| | ∞-Ter-Pinene | Cineol | Y-Ter Pinene | P–Cy–Mene | Terp-Ineolene | D-Lin-Alool | Terpin-Eol 4ol | Bcar-Yophe Llene | ∞-Ter-Pineol | Tymol | Carvacol | Total% |
| NE.NPK | 7.15 | 1.44 | 12.51 | 2.62 | 2.24 | 1.23 | 23.41 | 1.92 | 3.64 | 9.95 | 9.32 | 67.43 |
| NE.MO | 7.75 | 1.62 | 12.81 | 2.75 | 2.62 | 1.71 | 23.99 | 1.97 | 3.92 | 9.89 | 1.52 | 69.03 |
| NE.FA | 7.53 | 1.48 | 12.60 | 2.68 | 2.42 | 1.53 | 23.66 | 1.95 | 3.71 | 9.75 | 1.44 | 68.75 |
| GI.NPK | 7.75 | 1.10 | 12.79 | 2.72 | 2.64 | 1.52 | 23.89 | 1.85 | 3.75 | 9.87 | 1.70 | 69.58 |
| GI.MO | 7.92 | 1.18 | 12.91 | 2.82 | 2.86 | 1.76 | 23.93 | 1.92 | 3.89 | 9.91 | 1.87 | 71.60 |
| GI.FA | 7.89 | 1.77 | 12.99 | 2.83 | 2.71 | 1.60 | 23.95 | 1.90 | 3.81 | 9.97 | 1.62 | 71.04 |
| CH.NPK | 7.69 | 1.52 | 12.75 | 2.66 | 2.63 | 1.05 | 23.76 | 1.78 | 3.72 | 9.75 | 1.75 | 69.66 |
| CH.MO | 8.77 | 2.72 | 13.80 | 3.51 | 3.45 | 2.32 | 24.81 | 2.75 | 3.63 | 10.80 | 2.25 | 79.81 |
| CH.FA | 8.42 | 2.35 | 13.31 | 3.25 | 3.31 | 2.25 | 24.35 | 2.27 | 4.35 | 10.42 | 2.20 | 76.48 |
| YS.NPK | 7.80 | 1.72 | 12.82 | 2.71 | 2.60 | 2.55 | 23.85 | 1.88 | 3.81 | 9.86 | 1.72 | 71.32 |
| YS.MO | 8.35 | 2.18 | 13.32 | 3.15 | 3.21 | 3.32 | 24.35 | 2.27 | 4.19 | 10.25 | 2.35 | 70.94 |
| YS.FA | 8.25 | 2.11 | 13.25 | 3.12 | 3.18 | 3.27 | 24.11 | 2.15 | 4.12 | 10.17 | 2.23 | 75.86 |
| LSD1% | | | | | | | | | | | | 0.25 |

### 4.4.2. Total Phenolic (TPC)

TPC, 7.83 mg GAE/g. DLW. (the control was increased significantly as % of control (Table 3 and Figure 2) as the following:

CH.MO, CH.FA, and CH.NPK were 171, 159, and 97, respectively, and YS.MO, YS.FA, and YS.NPK were 156, 135, and 85, respectively. In addition, GI.MO, GI.FA, GI, and NPK were 122, 109, and 44, respectively. Furthermore, NE.MO, NE.FA, and NE NPK were 63, 51, and 0 (control = 7.83 mg GAE/g DLW).

### 4.4.3. Total Flavonoids Content (TFC)

TFC, 0.15 mg QE/g DLW (control) was increased significantly, as % of control (Table 3 and Figure 2) as the following:

**Table 3.** Marjoram total phenolic (TPC), flavonoid (TFC), flavonols (TFL), tannin (TAN) in response elicitation with gamma irradiation (GI), chitosan (CH) and yeast (YS) under chemical fertilizer (NPK) and organic fertilizer; Moringa (MO)and fulvic acid (FA) at 2 subsequent seasons (2019 and 2020).

| Application Treatment | TPC, mg GAE, g/DW | | | TFC, mg QE/DW | | | TFL, mg QE/gDW | | | TAN, mg TAE/gDW | | |
|---|---|---|---|---|---|---|---|---|---|---|---|---|
| | 2019 | 2020 | PM | 2019 | 2020 | PM | 2019 | 2020 | PM | 2019 | 2020 | PM |
| NE.NPK | 7.85 | 7.81 | 7.83(0) | 0.16 | 0.14 | 0.15(0) | 2.58 | 2.53 | 2.55(0) | 1.26 | 1.23 | 1.25(0) |
| NE.MO | 12.64 | 12.89 | 12.77(63) | 0.29 | 0.24 | 0.26(73) | 3.90 | 3.97 | 3.93(54) | 1.70 | 1.69 | 1.70(36) |
| NE.FA | 11.78 | 11.87 | 11.83(51) | 0.22 | 0.20 | 0.21(41) | 3.61 | 3.59 | 3.60(41) | 1.59 | 1.58 | 1.58(26) |
| GI.NPK | 11.15 | 11.32 | 11.24(44) | 0.24 | 0.22 | 0.23(53) | 4.26 | 4.28 | 4.27(67) | 1.74 | 1.75 | 1.75(40) |
| GI.MO | 17.27 | 17.57 | 17.42(122) | 0.33 | 0.29 | 0.31(106) | 4.82 | 4.83 | 4.82(89) | 2.03 | 2.03 | 2.03(62) |
| GI.FA | 16.25 | 16.48 | 16.37(109) | 0.28 | 0.25 | 0.26(73) | 4.59 | 4.60 | 4.59(80) | 1.85 | 1.86 | 1.86(49) |
| CH.NPK | 15.31 | 15.46 | 15.39(97) | 0.34 | 0.30 | 0.32(113) | 4.67 | 4.73 | 4.70(84) | 2.03 | 2.07 | 2.05(64) |
| CH.MO | 21.20 | 21.24 | 21.22(71) | 0.43 | 0.38 | 0.41(173) | 5.94 | 6.55 | 6.25(144) | 2.33 | 2.32 | 2.33(86) |
| CH.FA | 20.18 | 20.38 | 20.28(59) | 0.38 | 0.33 | 0.35(133) | 5.03 | 5.01 | 5.02(97) | 2.16 | 2.09 | 2.12(69) |
| YS.NPK | 14.37 | 14.60 | 14.49(85) | 0.29 | 0.26 | 0.28(87) | 4.44 | 4.43 | 4.44(74) | 1.92 | 1.89 | 1.91(53) |
| YS.MO | 19.70 | 20.15 | 19.93(125) | 0.35 | 0.32 | 0.34(127) | 5.39 | 5.87 | 5.63(121) | 2.17 | 2.14 | 2.16(73) |
| YS.FA | 18.13 | 18.67 | 18.40(135) | 0.32 | 0.29 | 0.30(100) | 4.82 | 4.78 | 4.80(88) | 2.03 | 1.99 | 2.01(61) |
| LSD1% | 0.07 | 0.08 | - | 0.04 | 0.03 | - | 0.05 | 0.06 | - | 0.06 | 0.07 | - |

PM: pooled mean for two Seasons (2019–2020) parentheses were % of control (NE > NPK) Values between parentheses are % of control.

Values of CH.MO, CH.FA, CH.NPK were 173, 133, and 113, respectively, and YS.MO, YS.FA, and YS.NPK were 127, 100, and 87 respectively. In addition, values of GI.MO, GI.FA, GI, NPK were 106, 73, and 53 respectively, and NE.MO, NE.FA, NE NPK were 73, 41, and 0 (control = 0.15 mg QE/g DLW).

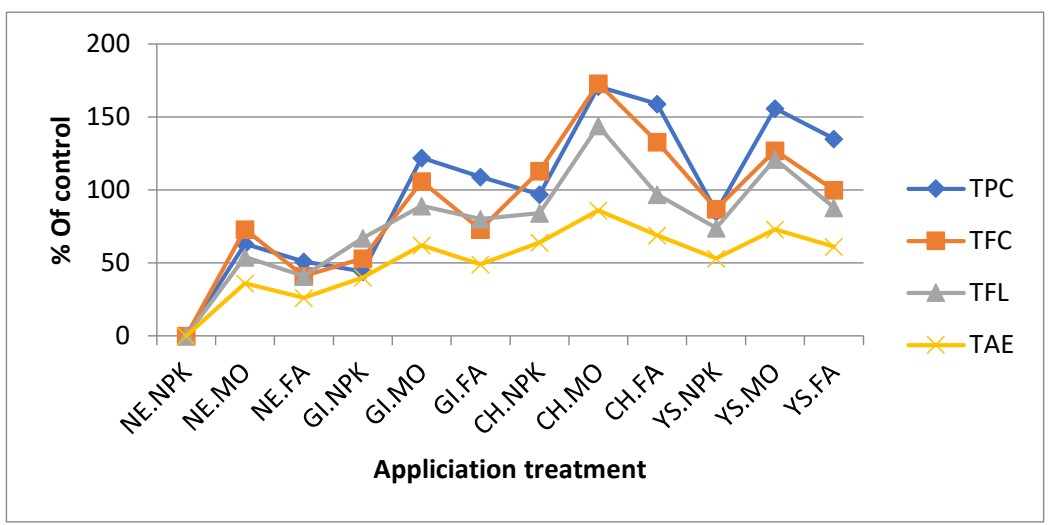

**Figure 2.** TPC, TFC, TFL, TAE, respectively.

4.4.4. Total Flavonols (TFL)

TFL, 2.55 mg QE/g. DLW, increased significantly, as % of control (Table 3 and Figure 2) as the following: values of CH.MO, CH.FA, and CH.NPK, 144, 97, and 84, respectively. Furthermore, YS.MO, YS.FA, YS. and NPK, 121, 88, and 74, respectively. In addition, values of GI.MO, GI.FA, GI, and NPK were 89, 80, and 67, respectively, and NE.MO, NE.FA, and NE NPK were 54, 41, and 0 (control = 2.55 mg QE/g DLW).

4.4.5. Tannin (TAN):

TAN;1.25 mgTAE/g. DLW, was increased significantly, as % of control, Table 3 and Figure 2 summarize the result, and show the following: CH.MO, CH.FA, and CH.NPK were 86, 69, and 64, respectively. In addition, values of YS.MO, YS.FA, and YS.NPK were 73, 61, and 53, respectively.

In addition, values of GI.MO, GI.FA, GI.NPK were 62, 49, and 40, respectively, and NE.MO, NE.FA, and NE.NPK were 36, 26, and 0 (control = 1.55 mg TAE/g DLW)

Elicitor application resulted in significant positive impacts for marjoram BSM production and quality (EO, TPC, TFC, TFL and TAN) as CH >YS > GI integrated with MO > FA > NPK.

Several researchers have supported our results [17,33] under agrochemical agriculture which impacts the environment preventing sustainable development [23].

Furthermore, under bio-fertilizer bio or organic fertilizers, organic, BSMs were increased [35,36].

The overall results manifested strong evidence that CH > YS > GI integrated with MO > FA > NPK could be considered as a reliable technological strategy to improve biomass and BSM (EO, TPC, TFC, TFL, TAN) production and quality of marjoram plants. The disappearance of any microbial diseases or pest infestation from the plant without using any pesticides such as micro-biocides is attributed to the following: (1) enhancing tolerance of plants to biotic and abiotic stresses [37] and leading to strong resistance to pests, microbial agents and nematodes [34]; (2) induced systemic resistance (ISR) prior to infection by regulating systemic resistance (ISR) prior to infection by regulating the expression genes involved in the production and accumulation of bioactive secondary metabolites (phytoalexins) which specific toxins characterized a broad spectrum of biomachrobiocide and biopesticides, making them less susceptible against microbial diseases and pest

infecting [38]. Furthermore, (3) chitosan (CH); exhibited strong antifungal and antibacterial [39] nematocidal, virucidal [40,41] and biopesticide attributes [42]. Moringa exhibited anti-fungal traits [43], and improved resistance to pests and microbial diseases [44,45].

## 5. Conclusions

The overall results show strong evidence for the potent of biotic elicitors chitosan > yeast > abiotic, gamma irradiation coupled with organic fertilizers moringa > fulvic acid > traditional agrochemical fertilizer NPK, as reliable CO-friendly solutions, significantly improving marjoram biomass, secondary metabolites production and quality without using agrochemical pesticides and/or microbicides. In addition, chitosan > yeast > gamma irradiation coupled with organic fertilizers exceeded their integration with chemical NPK fertilizer. This highlights the sustainable and reclusive development for marjoram under organic agriculture and could be an alternative to conventional agrochemical agriculture. This section is not mandatory but can be added to the manuscript if the discussion is unusually long or complex.

**Author Contributions:** All authors have equal contributions. All authors have read and agreed to the published version of the manuscript.

**Funding:** This research received no external funding.

**Institutional Review Board Statement:** Not applicable.

**Informed Consent Statement:** Not applicable.

**Data Availability Statement:** Not applicable.

**Conflicts of Interest:** The authors declare no conflict of interest.

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
