# Peer review of "Elicitation Promoability with Gamma Irradiation, Chitosan and Yeast to Perform Sustainable and Inclusive Development for Marjoram under Organic Agriculture"

_sustainability, doi:10.3390/su14159608_

Round 1
Reviewer 1 Report
Dear Authors,
In my opinion, the quality of the formal execution of the manuscript is not worthy of the standard of the journal. For this reason, I recommend a major revision of the manuscript according to the formal requirements of the scientific articles and according to the instructions for authors (of Sustainability journal).
Author Response
Thank you very much for your time and efforts in reviewing my manuscript, and I confirm that we reviewed the entire manuscript and made significant modifications to cover all formal requirements for scholarly articles consistent with the authors' instructions for Sustainability Journal. Please see the revised version with yellow highlights, I hope it will be accepted in the current form.

Reviewer 2 Report
i feel that your article could be reconsidered for publication should you be prepared to incorporate revisions.
After reading the article, I came away with the feeling that it needs serious revision.
1- Please digest more recent references in the introduction.
2- Please try to improve the objective to be clear.
3. Please check the similarity level, especially in the Methodology sections.
4. Please try to digest recent references in the Discussion section.
5. Please try to write paragraph about the contribution to the knowledge.
Author Response
I feel that your article could be reconsidered for publication should you be prepared to incorporate revisions. After reading the article, I came away with the feeling that it needs serious revision.
- Thank you very much for your time and your efforts in reviewing my manuscript and for your valuable comments. We have revised the manuscript completely to be suitable and acceptable for publication as recommended.
Point 1: Please digest more recent references in the introduction.
Response 1: The introduction has been provided with more recent references as recommended.
Point 2: Please try to improve the objective to be clear.
Response 2: The manuscript has been completely revised and modified to be suitable for publication with a clearer objective.
Point 3: Please check the similarity level, especially in the Methodology sections.
Response 3: Similarity was reduced to a particularly high degree in the methodology as recommended.
Point 4: Please try to digest recent references in the Discussion section.
Response 4: The manuscript has been completely modified and provided with more recent references to be suitable and acceptable for publication.
Point 5: Please try to write paragraph about the contribution to the knowledge.
Response 5: An introductory paragraph has been provided in the Introduction section topic to add a short background and a brief introduction to contribute knowledge.
Reviewer 3 Report
1. Keywords should be in alphabetical order.
2. Another correction should be done, which have been remarked in different color on the original manuscript.
3. References from 29 to 35 should be (in a form) like the other references, particularly the year of publication. Please see them.

Author Response
Point 1: Keywords should be in alphabetical order.
Response 1: Keywords are arranged alphabetically.
Point 2: Another correction should be done, which have been remarked in different color on the original manuscript.
Response 2: The manuscript has been completely revised and modified to be suitable for publication.
Point 3: References from 29 to 35 should be (in a form) like the other references, particularly the year of publication. Please see them.
Response 3: All references have been formatted to be appropriate for the journal's instructions.

Round 2
Reviewer 1 Report
Dear Authors,
Despite the modifications, the quality of the manuscript did not improve significantly.
In my opinion, the quality of the formal execution of the manuscript is not worthy of the standard of the journal.
Reviewer 2 Report
i think that the article can be accepted for publication